# A method to correct for the effect of blockage and wakes on power performance measurements

Alessandro Sebastiani[1], James Bleeg[2], and Alfredo Peña[1]

[1]DTU Wind and Energy Systems, Frederiksborgvej 399, 4000 Roskilde, Denmark
[2]DNV, One Linear Park, Avon St, Temple Quay, Bristol BS2 0PS, UK

**Correspondence:** Alessandro Sebastiani (aseb@dtu.dk)

**Abstract.**

Wind turbine power performance measurements often occur at the perimeter of a wind farm, where the wind flow is subject to blockage effects, which might impact the measured power performance. We perform Reynolds-averaged Navier-Stokes simulations of a wind farm with five rows of twenty turbines each, operating in a conventionally neutral boundary layer, to evaluate
whether the power performances measured for turbines in the upstream row would differ from that of a turbine operating in isolation under the same inflow conditions. We simulate the power performance measurements with both meteorological masts and nacelle-mounted lidars. Results show that blockage effects have an impact on the measured power performance of the wind farm turbines, with measured power coefficient varying more than 1% relative to what is measured for the isolated turbine. In this work, we propose a method to correct for the effect of blockage on power performance measurements, yielding a curve that
is more consistent with how power curves in energy yield analyses are defined and used, and thereby allowing for more useful comparisons between these curves. Our numerical results indicate that the correction method greatly reduces blockage-related variance and bias in the measured power curves. While flow modelling can be used to calculate the correction factors for actual power performance measurements in the field, we additionally show how some of the correction factors can be derived from lidar measurements. Finally, the numerical results suggest that the method could also be used to correct for the effect of wakes
on power performance measurements conducted on turbines located downstream of the leading row.

## 1 Introduction

Wind turbine power curve measurements play an important role in the wind industry. Manufacturers use them to better understand the performance of their fleet of operating turbines, and also to refine their power predictions for new, untested designs. Wind farm owners use on-site power performance measurements to determine whether their turbines are performing at a level
consistent with the predicted, theoretical power curves provided by the manufacturer. The vast majority of power performance measurements are conducted in wind farms for this purpose. Any assessment of discrepancies between actual wind farm energy production and the pre-construction estimate is not complete without verification of turbine power performance.

In an energy yield analysis, theoretical and warranted turbine power curves are the key link between the expected freestream wind resource and the predicted energy production of a planned wind farm. As such, theoretical power curves are traditionally

defined as functions of hub-height freestream wind speed. When running a power performance verification test, it is straight-
forward to measure the power; however, the corresponding freestream wind speed—i.e. the horizontal wind speed that would
prevail at the turbine location if the wind turbine was not there—is not a measurable quantity. Instead, power performance
measurement campaigns are designed to measure a wind speed that has traditionally been expected to be very close to what the
hub-height freestream wind speed would be if we could measure it. The IEC standard for power performance measurements

(IEC, 2017) requires the mast or lidar to measure between two and four rotor diameters (D) upstream of the test turbine, close
enough for the flow to be well correlated with conditions at the turbine, but far enough, ostensibly, for the influence of turbine
induction on the measured wind speed to be negligibly small. In addition, the measurement location and valid wind directions
are restricted to avoid upstream wakes. The IEC standard states the purpose of these requirements clearly (IEC, 2017): "The
WME (wind measurement equipment) shall not be influenced by the wind turbine under test. The wind turbine under test and

the WME shall not be influenced by neighbouring operating wind turbines."

Despite these restrictions, there is growing evidence that turbine-related disturbances materially influence power perfor-
mance measurements. The most compelling evidence involves field observations. Asimakopoulos et al. (2014) showed turbine-
induced velocity reductions up to 3.5D upstream of the rotor by measuring with lidars installed on both the nacelle and the
transition piece of an offshore wind turbine. Nacelle-mounted lidar measurements at eight different offshore wind farms re-

ported by Nygaard and Brink (2017) showed that the wind speeds measured 2.5D upstream of the test turbines were below
freestream, an average of 1.0% below according to their estimate. Based on this finding, they recommended applying an "in-
duction correction factor" when calculating energy yield using a measured power curve or similarly productive theoretical
curve. Using meteorological mast measurements taken before and after the start of operation at three onshore wind farms,
Bleeg et al. (2018) found that wind speeds measured 2D upstream the wind farms decreased by 3.4%, on average, relative to

wind speeds measured farther away after the turbines started operating. The observed slowdowns were well in excess of what
could be attributed to induction of a single turbine, which in part led to the conclusion that the other wind farm turbines also
contributed to these slowdowns. Based on additional analysis, they further concluded that wind farm blockage not only reduces
the wind speed upstream of the wind farm, but it also reduces the wind speed experienced by the turbines on the upstream
perimeter of the wind farm, causing them to generally produce less than they would operating in isolation. An analysis of

power performance measurements conducted in a row of five turbines, along with a complementary set of Reynolds-Averaged
Navier-Stokes (RANS) simulations, showed that wind farm blockage materially influences the measurements (Sebastiani et al.,
2022). Specifically, wind farm blockage appears to affect the ratio between the wind speeds at the mast location and the rotor.
Beyond field observations, there are also simulation-based studies (Allaerts and Meyers, 2017; Meyer Forsting et al., 2017;
Nishino and Draper, 2015; Strickland and Stevens, 2022; Bleeg and Montavon, 2022) and wind tunnel studies (Medici et al.,

2011; Ebenhoch et al., 2017; Segalini and Dahlberg, 2020; McTavish et al., 2015) that highlight turbine-related flow distur-
bances that likely affect power performance measurements.

The IEC standard explains how to correct for flow distortions caused by terrain, but there is nothing on how to correct
for flow disturbances/distortions caused by wind turbines. Although there is emerging recognition that turbine-induced flow
disturbances should be accounted for, the wind energy community at present lacks a generally accepted method to quantify the

impact of these flow disturbances and thereby correct for them. Specifically, although several models have been developed to account for blockage effects on turbine interaction loss (Nygaard et al., 2020; Branlard and Meyer Forsting, 2020; Segalini, 2021; Bleeg, 2020), but accounting for blockage effects on power performance measurements is still a rather unexplored topic.

Here, we propose a method to correct for the impact of turbine-related disturbances on power performance measurements. The methodology, which applies to both mast- and lidar-based measurements, is designed to yield power curves that are consistent with how theoretical curves are defined. After describing the correction method in detail, including the reasoning behind it, we test the method using RANS simulations of a notional wind farm. Finally, we use virtual nacelle lidar measurements to explore whether the correction can be completed, at least partly, using nacelle lidar measurements rather than flow simulations alone.

The work is organized as follows. In Sect. 2, the correction method is explained. In Sect. 3, the numerical model is presented with descriptions of the computational fluid dynamics (CFD) model (Sect. 3.1), the simulation set-up (Sect. 3.2) and the virtual lidar measurements (Sect. 3.3). Results from power performance measurements conducted on the first upstream row of a wind farm are shown in Sect. 4, while in Sect. 5 we show how short-range nacelle lidar measurements can be used to apply the correction method. In Sect. 6, the correction method is applied to all turbines in the wind farm, including downstream waked turbines. Finally, discussion and conclusions are presented in Sects. 7 and 8, respectively.

## 2 Correction method

Common practice, when estimating the energy yield of a planned wind farm, is to combine the expected freestream wind resource at each turbine location with the manufacturer-provided theoretical power curve to calculate the so-called gross energy. This is the total of the energy that each turbine would produce absent the presence of the other wind turbines and other loss sources. The net energy is obtained after turbine interaction and other losses are accounted for. Thus, the power curve used in an energy yield analysis should faithfully represent the power production of the turbine as function of freestream wind speed when the turbine is operating in isolation. We refer to this power curve definition as a freestream power curve, $P(U_\infty)$.

A power curve measured according to IEC standards, $P(U_{\mathrm{mast}})$, is not a freestream power curve as defined above. The test turbine affects the measured wind speed via induction, and the other wind farm turbines affect the relationship between that wind speed and conditions at the rotor face, via blockage and sometimes wakes. The impact of these effects on the measured power curve should be quantified and corrected. The objective of the correction method described in this section is to convert the measured power curve to a freestream curve that can defensibly be compared with a theoretical power curve. In our approach, we only alter the wind speed column in the tabular power curve. Specifically, for a given measured power vs. wind speed pair in the table, we correct to the freestream mean wind speed that would prevail if the test turbine were producing the same amount of power while operating in isolation. The correction can be thought of a two-step process:

– Convert the measured curve to what would be measured if the test turbine were operating in isolation and producing the same amount of power measured in the test.

– Correct for the impact of induction from the isolated turbine on the mast wind speed.

When measuring the power performance of a turbine inside a wind farm, the measured power curve $P^{\mathrm{WF}} = P(U_{\mathrm{mast}}^{\mathrm{WF}})$ differs from the power curve that would be measured if the turbine were operating in isolated condition $P^{\mathrm{I}} = P(U_{\mathrm{mast}}^{\mathrm{I}})$, since both $P$ and $U_{\mathrm{mast}}$ are affected by surrounding turbines. Consequently, since both power and wind speed are different ($P^{\mathrm{WF}} \neq P^{\mathrm{I}}$ and $U_{\mathrm{mast}}^{\mathrm{WF}} \neq U_{\mathrm{mast}}^{\mathrm{I}}$), both $U_{\mathrm{mast}}^{\mathrm{WF}}$ and $P^{\mathrm{WF}}$ should be corrected in order to retrieve the power performance of the isolated turbine from wind farm measurements. However, if we consider the case with the isolated and the wind farm turbines producing the same amount of power $P = P^{\mathrm{I}} = P^{\mathrm{WF}}$, we would only need to correct the wind speed measurement, retrieving $U_{\mathrm{mast}}^{\mathrm{I}}$ from $U_{\mathrm{mast}}^{\mathrm{WF}}$.

Although wind turbine power is commonly formulated as a function of freestream wind speed, it is more directly a function of the velocity across the rotor face, which, along with air density and rotor speed, determines the aerodynamic loads on the blades. In this paper, we use the average axial velocity across the rotor face as a power-equivalent wind speed, $U_{\mathrm{disk}}$. In other words, if two turbines experience the same $U_{\mathrm{disk}}$, they are assumed to produce the same amount of power regardless of the respective $U_{\mathrm{mast}}$ and $U_{\infty}$ values. Thus, if $P^{\mathrm{I}} = P^{\mathrm{WF}}$, then $(U_{\mathrm{disk}}^{\mathrm{I}}/U_{\mathrm{disk}}^{\mathrm{WF}}) = 1$, and $U_{\mathrm{mast}}^{\mathrm{I}}$ can be reconstructed from $U_{\mathrm{mast}}^{\mathrm{WF}}$ as

$$U_{\mathrm{mast}}^{\mathrm{I,rec}} = U_{\mathrm{mast}}^{\mathrm{WF}} \left( \frac{U_{\mathrm{disk}}}{U_{\mathrm{mast}}} \right)^{\mathrm{WF}} \left( \frac{U_{\mathrm{mast}}}{U_{\mathrm{disk}}} \right)^{\mathrm{I}}, \tag{1}$$

where $U_{\mathrm{mast}}^{\mathrm{I,rec}}$ is the reconstructed velocity at the mast of the isolated turbine, $U_{\mathrm{mast}}^{\mathrm{WF}}$ is the velocity measured at the wind-farm mast, and the ratios $(U_{\mathrm{disk}}/U_{\mathrm{mast}})^{\mathrm{WF}}$ and $(U_{\mathrm{mast}}/U_{\mathrm{disk}})^{\mathrm{I}}$ are computed from numerical simulations of the wind farm and the isolated turbine, respectively.

The ratio $(U_{\mathrm{mast}}/U_{\mathrm{disk}})^{I}$ relates to the turbine blockage/induction and can be assumed to be nearly constant with small changes in wind speed over the plateau of the thrust-coefficient curve $C_T = C_T(U_{\infty})$. Therefore, Eq. (1) is still valid in the case of $U_{\mathrm{disk}}^{\mathrm{WF}} \neq U_{\mathrm{disk}}^{\mathrm{I}}$, as long as the turbine is operating at nearly the same thrust coefficient $C_T = C_T^{\mathrm{WF}} = C_T^{\mathrm{I}}$ and at a similar wind speed.

When $U_{\mathrm{mast}}^{\mathrm{I}}$ is retrieved at a distance of 2D upstream of the rotor, it might be affected by turbine blockage. Therefore, a similar approach as that used to derive Eq. (1) can be applied to reconstruct the freestream velocity:

$$U_{\infty}^{\mathrm{rec}} = U_{\mathrm{mast}}^{\mathrm{I,rec}} \left( \frac{U_{\infty}}{U_{\mathrm{mast}}} \right)^{\mathrm{I}}, \tag{2}$$

where $U_{\mathrm{mast}}^{\mathrm{I,rec}}$ is given by Eq. (1) and $(U_{\infty}/U_{\mathrm{mast}})^{\mathrm{I}}$ is computed from simulations of both the isolated turbine and the undisturbed free flow.

Some variations are expected for both $(U_{\mathrm{mast}}/U_{\mathrm{disk}})^{\mathrm{I}}$ and $(U_{\infty}/U_{\mathrm{mast}})^{\mathrm{I}}$ depending on the wind direction, as the degree of blockage at the mast depends on turbine yaw. Therefore, we also simulate IEC-compliant measurements with a 2-beam nacelle-mounted lidar, which yaws with the turbine. In those cases, we refer to the IEC wind speed measurement as either $U_{\mathrm{lidar}}^{\mathrm{I}}$ or $U_{\mathrm{lidar}}^{\mathrm{WF}}$.

## 3 Numerical model

### 3.1 CFD model

The numerical simulations are run using a CFD model based on STAR-CCM+, a general-purpose CFD software. The model solves the steady-state RANS equations along with a transport equation for potential temperature. The turbulence model is standard $k - \epsilon$ with modified coefficients. Buoyancy effects are captured through the addition of a gravity term in the vertical momentum equation, which is formulated using a shallow Boussinesq approximation. Buoyancy source terms are also included in the turbulence equations. More details about the flow model may be found in Bleeg et al. (2015b) and Bleeg et al. (2015a).

The turbines are represented via an actuator disk model. The disk volumes are discretised with cubic mesh cells with edge lengths equal to 5% of the rotor diameter (20 cells across the rotor diameter and 5 cells across the disk thickness). The axial and tangential body forces applied to the disk are modelled as a function of the disk-averaged axial velocity at the rotor face when the turbine is operating ($U_{\mathrm{disk}}$). Since manufacturer-provided curves for power and thrust coefficient ($C_T$) are functions of freestream wind speed ($U_\infty$), the curves used in the simulations need to be reformulated to be functions of $U_{\mathrm{disk}}$. The conversion of the manufacturer-provided $C_T$ and power curves follows a procedure similar to that of van der Laan et al. (2015). The procedure involves running a series of single-turbine simulations, each corresponding to a different hub-height wind speed. In these simulations, the $U_\infty$ values are known, and actuator disk forces are thereby set according to theoretical curves specified as functions of $U_\infty$. After each simulation finishes, we record $U_{\mathrm{disk}}$. The outcome of the conversion is a set of curves ($P'(U_{\mathrm{disk}})$, $C_T'(U_{\mathrm{disk}})$, and rotor speed) specified as a function of $U_{\mathrm{disk}}$.

All simulations correspond to a conventionally neutral boundary layer with a thickness of approximately 1000 m. The maximum potential gradient in the capping inversion is 10 K/km, and the free atmosphere above is stably stratified with a vertical potential temperature gradient of 3.3 K/km. In this numerical experiment, three types of simulations are run: full wind farm, turbines in isolation, and freestream. As the labels imply, the full wind farm simulations include all the wind turbines, the isolated turbine simulations only include one turbine, and the freestream simulations have no turbines/actuator disks. The three types of simulations are run with the same mesh and boundary conditions.

### 3.2 Simulation set-up

We perform RANS simulations of a wind farm with five rows of 20 turbines, as shown in Fig. 1-(a). The turbines have a rated power of 3.45 MW, rotor diameter of 136 m and hub height of 98 m. They are distributed with spacings of 3D and 10D along the $x$ and $y$ directions, respectively. We simulate five different wind directions, covering the sector from $-45°$ to $+45°$ with respect to the orthogonal wind direction $\theta = 0°$ as shown in Fig. 1-(a). The turbines are numbered starting from the most downwind row, so that turbines from T81 to T100 are wake-free for all the simulated wind directions.

Simulations are also performed with a single turbine operating within the same domain and under the same free-flow conditions of the wind farm. We simulate four different single-turbine cases in order to evaluate whether numerical effects cause different results for the isolated turbine when this is placed at different locations. We simulate the single turbine at the locations of T28, T81, T92 and T100 and, although not shown, we find that the results are independent of the location of the isolated

turbine. Therefore, in the following analysis, when we refer to the case with the isolated turbine, we point at the isolated turbine at T92, as shown in Fig. 1-(b).

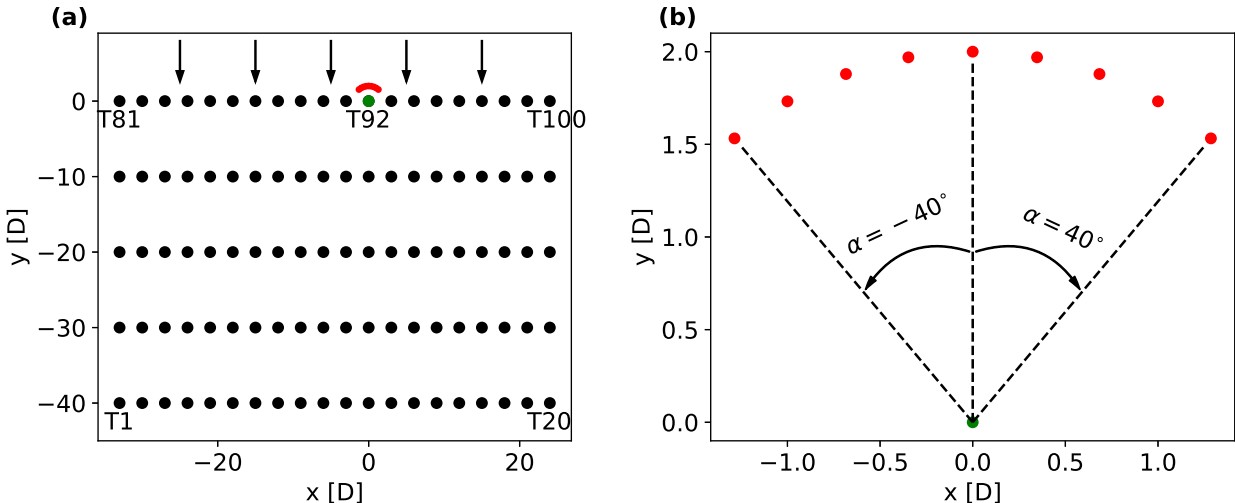

**Figure 1.** Illustrations of the wind farm layout $(a)$ and the isolated turbine $(b)$, with both wind turbine (*black circles*) and mast (*red circles*) locations. The arrows in $(a)$ show the $\theta = 0°$ wind direction.

To test the correction method, i.e., Eqs. (1) and (2), we extract the wind speed at hub height in front of all the first-row turbines in the full wind farm simulation and in front of the the isolated turbine simulation. To test all the possible IEC-compliant wind speed measurements, we simulate nine masts located on the 2D-radius circle around each turbine and distributed every $10°$
from $\alpha = -40°$ to $\alpha = 40°$ relatively to the north, as shown in Fig. 1-(b). It should be noted that, according to the IEC standard, the available sector for power performance tests for the full wind farm simulation would be larger than $[-40°, +40°]$ for the locations of T81 and T100, as there are no neighbouring turbines on one or both sides of these locations. However, to keep consistency in the comparison between the 20 upstream turbines and the isolated turbine, we consider the same sector of $-40°$ to $40°$ for all these turbine locations.
We aim to simulate five wind directions regularly distributed over the $[-45°, +45°]$ interval. However, the simulated flow field is characterized by vertical veer due to the combination of surface friction and Coriolis force, so the wind direction varies around $4°$ from bottom to top of the rotor swept area, as shown in Fig. 2-(b), with wind directions at hub height of $-46°$, $-23°$, $-1°$, $20°$ and $44°$. Additionally, Fig. 2-(a) shows the vertical velocity profiles, which are all characterized by a horizontal wind speed of around 7.1 m/s at hub height, with variations from $\sim$5.9 to $\sim$7.8 m/s across the rotor swept area. The wind speed was
chosen so that the all the simulated turbines operate on the plateau of the $C_T$ curve.

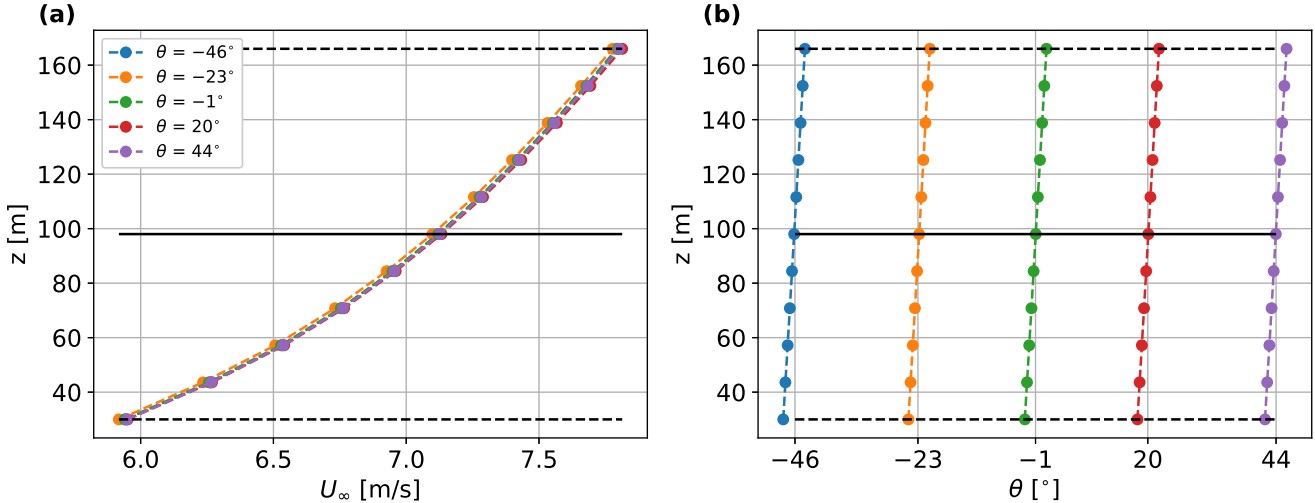

**Figure 2.** Vertical profiles of the horizontal wind speed ($a$) and wind direction ($b$) extracted at the location of T92 from the freestream simulations. Black lines indicate hub height (*continuous*) and rotor tips (*dashed*).

### 3.3 Virtual lidar measurements

The correction method is based on the combination of measurements ($U_{\mathrm{mast}}$ or $U_{\mathrm{lidar}}$) with the numerically computed value of $U_{\mathrm{disk}}$, which is hard to estimate out in the field. Therefore, we investigate whether $U_{\mathrm{disk}}$ can be replaced with a measurable velocity quantity: we simulate short-range nacelle lidar measurements in the induction zone and derive a velocity quantity
$U_{\mathrm{disk,lidar}}$ that is tested as a proxy for $U_{\mathrm{disk}}$ in Eq. (1). Furthermore, we simulate IEC-compliant nacelle lidar measurements to evaluate the performance of the correction method when replacing $U_{\mathrm{mast}}$ with $U_{\mathrm{lidar}}$ in Eq. (1).

We retrieve the IEC-compliant wind speed measurements with a 2-beam nacelle-mounted lidar measuring at 2D upstream of the rotor with a half-opening angle $\varphi = 15°$. Additionally, as shown in Fig. 3, we retrieve wind speed values at 0.5D upstream of the rotor with four different nacelle lidars: the same 2-beam lidar used to measure at 2D; a 4-beam lidar with $\varphi = 18°$ and
180 the measurement points at the four vertices of a square; a 50-beam circularly scanning lidar with $\varphi = 15°$; and an additional 50-beam ideal lidar that scans along the circular pattern of radius equal to three quarters of the rotor radius. The choice of the 50-beam ideal lidar scanning pattern is based on the work by Sebastiani et al. (2023), who showed that, among several circular scanning patterns, the one scanning at around a three quarter of the radius provided the highest accuracy in power prediction.

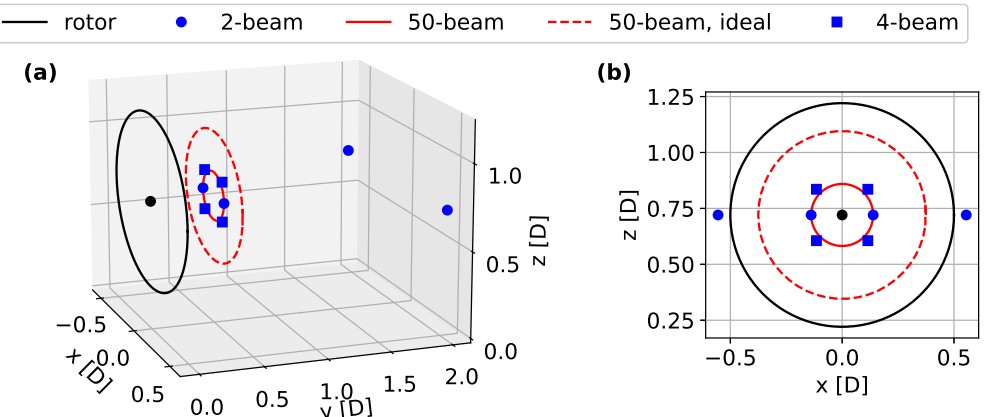

**Figure 3.** Illustrations of the rotor and lidar measurement points at both 2 and 0.5D with three- and two-dimensional views in $(a)$ and $(b)$, respectively. The *black circle* indicates the location of the lidar

We assume horizontal homogeneity of the flow field to reconstruct the horizontal wind speed at hub height from the 2-beam lidar measurements by inverting the linear system

$$\begin{pmatrix} n_x^1 & n_y^1 \\ n_x^2 & n_y^2 \end{pmatrix} \cdot \begin{pmatrix} u_x \\ u_y \end{pmatrix} = \begin{pmatrix} v_r^1 \\ v_r^2 \end{pmatrix}, \tag{3}$$

where $n_j^i$ is the $j$-th component of the unit vector $n^i$ indicating the direction of the $i$-th beam, $v_r^i$ is the radial velocity retrieved from the $i$-th beam and $u_j$ is the $j$-th component of the horizontal wind velocity, whose magnitude is $U_{\text{lidar}} = \sqrt{u_x^2 + u_y^2}$. Under wake-free conditions, the assumption of horizontal homogeneity is reliable out of the induction zone, while close to the rotor, the flow field is characterized by a strong velocity gradient in the axial direction. However, since the two lidar beams measure at the same distance from the rotor, it is reasonable to assume horizontal homogeneity between the two measurement points. We do not simulate the lidar probe volume, which is important for turbulence estimations using lidars (Peña et al., 2017; Fu et al., 2022). Therefore, the radial velocities are retrieved as point measurements with a three-dimensional linear interpolation from the flow solution.

When using lidars with more than 2 beams, i.e., the two 50-beam and the 4-beam, we neglect both the lateral and vertical components of the wind speed vector by assuming $u_x = u_z = 0$ m/s, so that the horizontal wind speed at each beam location is retrieved as $u_y = v_r/n_y$. Then, the lidar-estimated disk velocity is obtained as the mean of the beam measurements: $U_{\text{disk,lidar}} = 1/n_{\text{beam}} \sum_{i=1}^{n_{\text{beam}}} u_y^i$. When using the 2-beam lidar focused at 0.5D, the horizontal wind speed at hub height retrieved through Eq. (3) is used as $U_{\text{disk,lidar}}$.

## 4 Power performance measurement of the first-row turbines

Wind farm blockage affects the flow upstream of the wind farm, impacting the velocity relative to the flow upstream of the isolated turbine. Figure 4-(a) shows the difference between the wind speed $U^{\mathrm{I}}_{\mathrm{mast}}$ measured in front of the isolated turbine and the wind speed $U^{\mathrm{WF}}_{\mathrm{mast}}$ measured in front of the $i^{th}$ wind-farm turbine for the same $j^{th}$ wind direction:

$$\Delta U_{ij} = 100 \, \frac{U^{\mathrm{WF}}_{\mathrm{mast}}(T_i, \theta_j) - U^{\mathrm{I}}_{\mathrm{mast}}(\theta_j)}{U^{\mathrm{I}}_{\mathrm{mast}}(\theta_j)}. \tag{4}$$

The error bars of Fig. 4-(a) indicate mean and standard deviations associated with the computations based on the 9 mast locations. For most of the first-row turbines, the measured wind speed is lower than that of the isolated case for all the simulated wind directions, with velocity reductions sometimes more than 3% in the centre of the row. However, in cases of highly skewed inflow, the wind speed is increased around the most downwind turbines. For $\theta = 44°$, $U^{\mathrm{WF}}_{\mathrm{mast}}$ at T81 is $\approx$1.5% higher than $U^{\mathrm{I}}_{\mathrm{mast}}$. The same trend of wind speed variations for a skewed inflow was found by Sebastiani et al. (2022) for a single row of wind turbines, where the downstream turbines are in the speed-up region formed at the edge of the wakes from the upstream turbines (Meyer Forsting et al., 2017). In our case, due to the size of the wind farm, the wind speed increase might be also due to the speed-up at the edge of the wind-farm induction region. The asymmetry in global-blockage effect between cases with almost symmetric inflow angles, such as $-46°$ and $44°$, is probably due to the asymmetry introduced by the vertical wind veer and wake rotation.

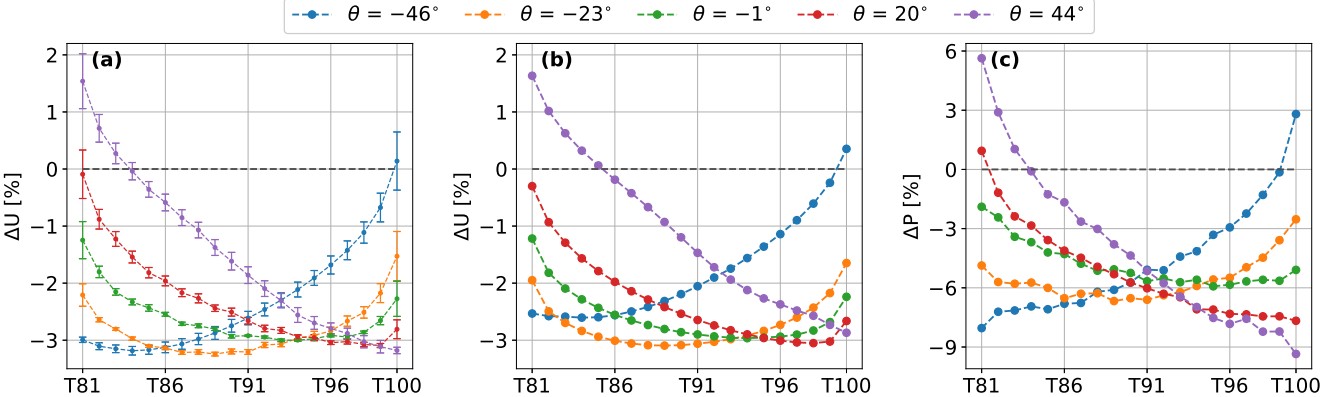

**Figure 4.** Variations in wind speed $(a, b)$ and power output $(c)$ relatively to the isolated turbine for all the simulated wind directions, when measuring the wind speed at 2D in front of the turbines with either masts $(a)$ or the 2-beam lidar $(b)$. Error bars in $(a)$ indicate the standard deviations related to the variation of $\alpha$.

The blockage-induced velocity variations do not change much when replacing the masts with a 2-beam nacelle-mounted lidar, as shown in Fig. 4-(b). However, the nacelle lidars measure the wind speed along the rotor axis irrespective of $\theta$, removing the variation associated with $\alpha$ and hence the error bars.

Similarly to the wind speed variations shown in Fig.s 4-(a) and (b), Fig. 4-(c) shows the power deviations of the first-row turbines relatively to the isolated turbine:

$$\Delta P_{ij} = 100 \, \frac{P^{\mathrm{WF}}(T_i, \theta_j) - P^{\mathrm{I}}(\theta_j)}{P^{\mathrm{I}}(\theta_j)}. \tag{5}$$

Since the power output is related to the velocity to the power of 3, power variations are larger in magnitude than the velocity ones, with variations from $-9.4\%$ to $+5.6\%$ with respect to the isolated turbines. Additionally, the largest power losses are not found for the central turbines as for the wind speed, but for the most upstream turbines in the case of strongly skewed inflows, i.e., T81 for $\theta = -46°$ and T100 for $\theta = 44°$.

Since $U_{\mathrm{mast}}^{\mathrm{WF}}$ and $P$ are not perfectly correlated, their blockage-induced variations cause uncertainty in the power curve, as shown in Fig. 5-(a), which shows the power output from the first-row turbines against the wind speed measured by their masts for all simulated wind directions $\theta$ and mast locations $\alpha$. The scatter in Fig. 5-(a) shows that the relation between the power output and the measured wind speed varies for different wind directions and mast locations. When applying the correction in Eq. (1) to $U_{\mathrm{mast}}^{\mathrm{WF}}$, the scatter in the power curve does not decrease much, as shown in Fig. (5)-(b). On the other hand, when 230 further correcting $U_{\mathrm{mast}}^{\mathrm{I,rec}}$ with Eq. (2), the scatter in the power curve decreases substantially to a much lower level as shown in Fig. 5-(c).

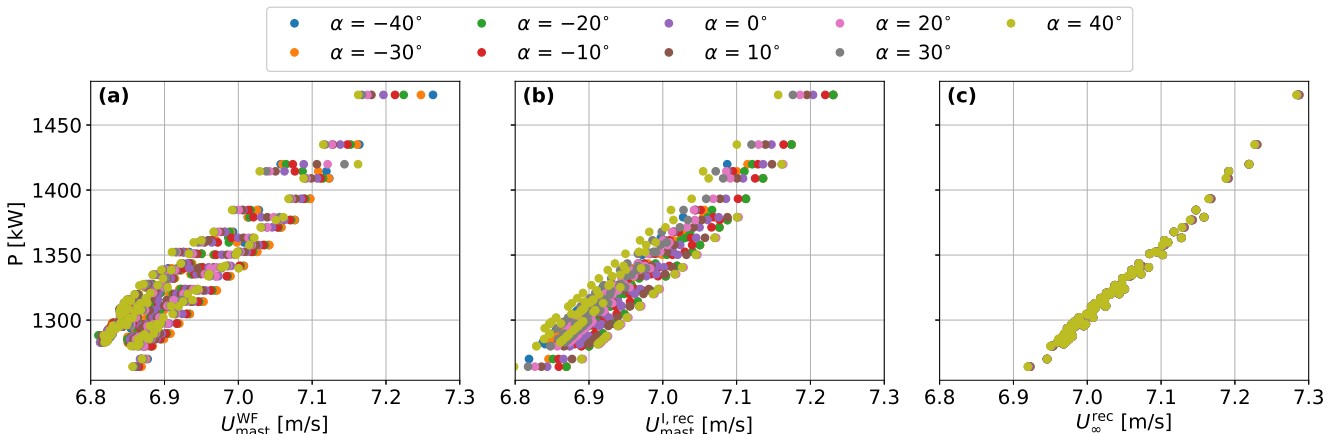

**Figure 5.** Power output against the mast-measured wind speed of all the first-row turbines for all wind directions and mast locations. $(a)$: no correction applied on the measured wind speed. $(b)$: wind speed corrected with Eq. (1). $(c)$: wind speed corrected with Eq. (2).

The scatter shown in Fig. 5-(a) is not only due to wind farm blockage, but also to the induction/blockage of the test turbine, whose effect is not accounted for using Eq. (1), which reconstructs the wind speed that would be measured around the isolated turbine that is producing the same amount of power as the wind-farm turbine. Since the blockage-induced velocity field is not 235 spatially uniform, the ratio $(U_{\mathrm{mast}}/U_{\mathrm{disk}})^{\mathrm{I}}$ varies with both $\theta$ and $\alpha$.

When applying Eq. (2), we are correcting for the induction of the test turbine, relating the power output to the freestream velocity that would be measured at the isolated turbine location if the turbine was not there. The power curves retrieved from

different masts collapse onto each other, as the freestream velocity does not vary substantially with either $\alpha$ or $\theta$ due to the nearly homogeneous velocity field.

Figure 6 shows scatter plots of the power output against the lidar-retrieved wind speeds for the first-row turbines. When using nacelle lidars, $\alpha$ can be disregarded and the power performance variations are due to the turbine location and wind direction only. The scatter is almost completely reduced by using Eq. (1), as nacelle lidars measure the wind along the rotor axis regardless of $\theta$ so that the measurements are equally affected by turbine blockage for different values of $\theta$. When using Eq. (2) to correct for turbine blockage, the scatter does not decrease and the only effect is the shift towards higher velocity values.

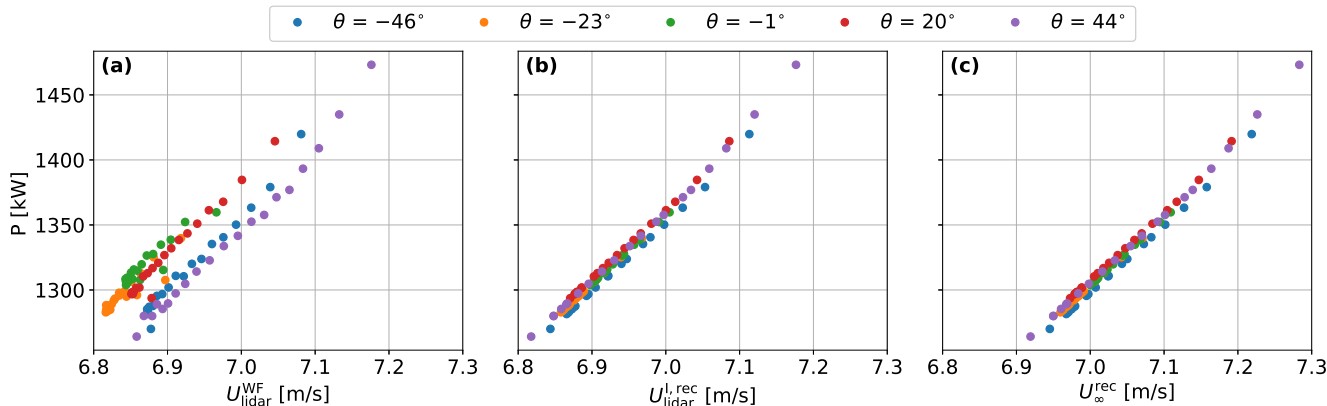

**Figure 6.** Power output against the lidar-measured wind speed of all the first-row turbines for all wind directions. ($a$): no correction applied on the measured wind speed. ($b$): wind speed corrected with Eq. (1). ($c$): wind speed further corrected with Eq. (2).

When measuring power curves with nacelle lidars, Eq. (1) can be used to correct wind-farm effects and retrieve the power performance of the isolated turbine as a function of measured wind speed. However, Eq. (2) is still needed to get the power output as function of the freestream velocity and to avoid an overestimation of the power performance, as it can be noticed in Fig. 6-(c), where power values are shifted to higher wind speed values compared to Fig. 6-(b).

Figure 7 shows the distributions of the $C_P$ values estimated using the different wind speed definitions. When looking at 250 the $C_P$ of the isolated turbine, we notice lower variation using lidar than mast measurements, as we avoid dependencies on $\alpha$ variations, so that the spread of the $C_P$ values is lower when using $U_{\mathrm{lidar}}^{\mathrm{WF}}$, $U_{\mathrm{lidar}}^{\mathrm{I,rec}}$ and $U_{\mathrm{lidar}}^{\mathrm{I}}$ than with $U_{\mathrm{mast}}^{\mathrm{WF}}$, $U_{\mathrm{mast}}^{\mathrm{I,rec}}$ and $U_{\mathrm{mast}}^{\mathrm{I}}$, respectively. If we assume $C_P = C_P(U_{\mathrm{lidar}}^{\mathrm{I}})$ as the reference value, the $C_P$ estimation is both inaccurate and imprecise when using either $U_{\mathrm{lidar}}^{\mathrm{WF}}$ or $U_{\mathrm{mast}}^{\mathrm{WF}}$. A variation of more than 6% is observed among the $C_P$ values estimated with $U_{\mathrm{mast}}^{\mathrm{WF}}$. The $C_P$ mean values are 1.5% and 1.4% higher than $C_P(U_{\mathrm{lidar}}^{\mathrm{I}})$ for $U_{\mathrm{mast}}^{\mathrm{WF}}$ and $U_{\mathrm{lidar}}^{\mathrm{WF}}$, respectively, whereas the interquartile range 255 (IQR) is 600% and 700% higher for $U_{\mathrm{mast}}^{\mathrm{WF}}$ and $U_{\mathrm{lidar}}^{\mathrm{WF}}$, respectively. By correcting with Eq. (1), the term $U_{\mathrm{mast}}^{\mathrm{I,rec}}$ provides higher accuracy than $U_{\mathrm{mast}}^{\mathrm{WF}}$ with both median and mean values closer to the reference, but the values are still highly spread due to the variations in $(U_{\mathrm{mast}}/U_{\mathrm{disk}})^{\mathrm{I}}$. On the other hand, we observe both an increase in accuracy and reduction in the spread when using $U_{\mathrm{lidar}}^{\mathrm{I,rec}}$, with differences with the reference of 0.4% and 19.8% for the mean value and IQR, respectively. However,

without applying Eq. (2), the $C_P$ values are not an accurate estimation of the power performance as $C_P = C_P(U_\infty)$. As shown in Fig. 7, the $C_P$ is overestimated relative to $C_P(U_\infty)$ by 4.1% and 4.5% when using $U_{\mathrm{mast}}^{\mathrm{I}}$ and $U_{\mathrm{lidar}}^{\mathrm{I}}$, respectively. On the other hand, the $C_P$ estimation is very accurate when using Eq. (2), with deviations of 0.4% from $C_P(U_\infty)$ for both $U_{\mathrm{mast}}^{\infty,\mathrm{rec}}$ and $U_{\mathrm{lidar}}^{\infty,\mathrm{rec}}$.

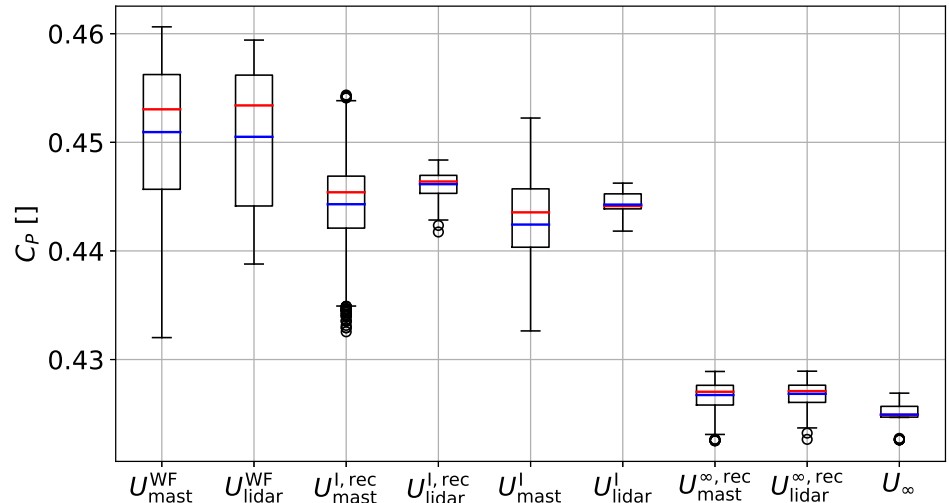

**Figure 7.** Box plots of the $C_P$ distributions for all wind directions and mast positions for the first-row turbines ($U_{\mathrm{mast}}^{\mathrm{WF}}$, $U_{\mathrm{lidar}}^{\mathrm{WF}}$, $U_{\mathrm{mast}}^{\mathrm{I,rec}}$, $U_{\mathrm{lidar}}^{\mathrm{I,rec}}$, $U_{\mathrm{mast}}^{\infty,\mathrm{rec}}$, $U_{\mathrm{lidar}}^{\infty,\mathrm{rec}}$) and for the isolated turbine ($U_{\mathrm{mast}}^{\mathrm{I}}$, $U_{\mathrm{lidar}}^{\mathrm{I}}$, $U_\infty$). Box plot features: quartiles $q1$ and $q3$ (*box limits*); lowest and highest values within $[q1 - 2\,\mathrm{IQR}, q3 + 2\,\mathrm{IQR}]$ (*whiskers*); values outside the range $[q1 - 2\,\mathrm{IQR}, q3 + 2\,\mathrm{IQR}]$ are shown as outliers (*circles*); median (*red line*) and mean (*blue line*).

It should be noted that the overestimation of $C_P$ observed with both $U_{\mathrm{mast}}^{\mathrm{WF}}$ and $U_{\mathrm{lidar}}^{\mathrm{WF}}$ might have strong implications on the accuracy of AEP estimations. Additionally, the wind speeds corresponding to the high $C_T$ values assumed in this work are usually among the most frequent wind speed values at typical wind farm sites (Hasager et al., 2006). Therefore, the results in Figs. 5, 6 and 7 show the need to correct for the effect of blockage on power performance measurements.

## 5   Lidar-based estimation of the disk velocity

We also investigate whether short-range nacelle lidar measurements can be used to replace the numerically-estimated $U_{\mathrm{disk}}$ in Eq. (1). In order to assess the correlation between $U_{\mathrm{disk}}$ and short-range nacelle lidar measurements, we simulate the 2-beam lidar focused at 19 different distances from the rotor, as shown in Fig. 8-(a). Specifically, we simulate measurements from the rotor plane up to 1.875D, and compute $U_{\mathrm{disk,lidar}}$ from the radial velocities of the two beams at each distance. We then show the correlation between $U_{\mathrm{disk,lidar}}$ and $U_{\mathrm{disk}}$ by implementing least-square linear regressions using the $(U_{\mathrm{disk,lidar}}, U_{\mathrm{disk}})$ values from all the 20 upstream turbines. The coefficients of determination $R^2$ of the regressions are shown in Fig. 9.

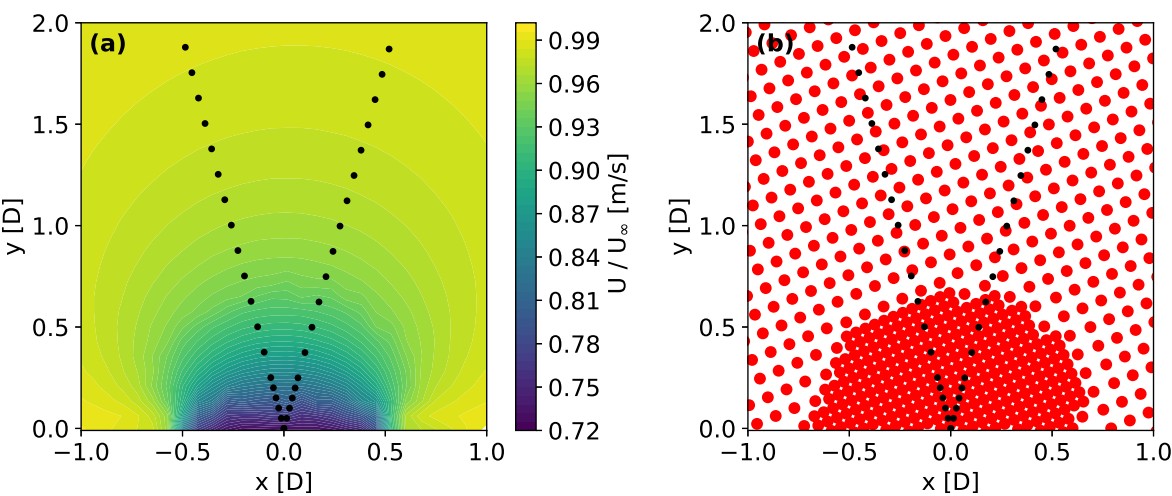

**Figure 8.** Normalized velocity field at hub height in front of the isolated turbine $(a)$ and grid discretization within the same area $(b)$.

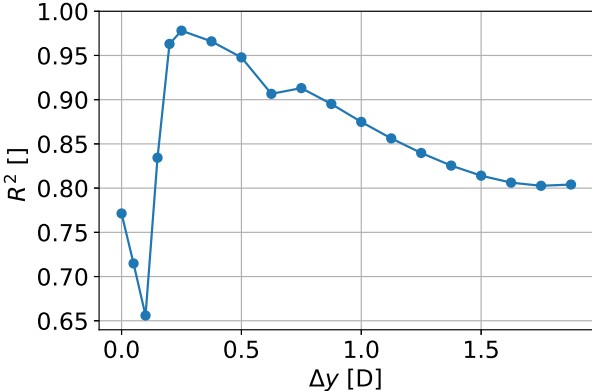

**Figure 9.** Variation of the coefficient of determination $R^2$ of the least-square linear regression between $U_{\mathrm{disk}}$ and $U_{\mathrm{disk,lidar}}$ for several upstream distances.

As it can be noted in Fig. 9, $R^2$ is low when measuring closer than 0.2D to the rotor and reaches its maximum at 0.25D, with a smooth decreasing trend for further distances. The value at $\Delta y = 0.625$ D appears as an outlier due to numerical biases because the focus point of the beams is at the edge of the highly discretized region (Fig. 8-b). The correlation between $U_{\text{disk}}$ and $U_{\text{disk,lidar}}$ decreases very close to the rotor. This is due to a combination of discretisation and interpolation errors. As shown in Fig. 8-(a), the flow field close to the rotor ($y \lesssim 0.2$D) is not as smooth as that far from the rotor. Strong velocity gradients near the rotor, caused by the applied turbine forces, increase discretisation and interpolation errors in this region.

We use the distance of $\Delta y = 0.5$D for testing the correction method based on the work by Troldborg and Meyer Forsting (2017), who showed that the induction zone is self-similar beyond 0.5D upstream of the rotor, i.e., that the induced velocity field is only function of the total $C_T$ with no dependency on the distribution of loads across the rotor. When measuring $U_{\text{disk,lidar}}$ at 0.5D, the results are representative of all wind turbine rotors, while using closer measurements might provide results which are representative of the simulated rotor only.

## 6  Power performance measurements in wakes

The correction method is not limited to blockage effects. In theory, it can be used to correct for any turbine-related disturbances, including wakes. Figure 10 shows the relation between wind speed and power output for all the 100 turbines in the farm and for all the simulated wind directions. We only consider nacelle lidars for power performance measurements of turbines T1, ..., T80. As shown in Fig. 10-(a), the power output is very poorly correlated with the hub-height wind speed measured at 2D in front of the rotor, as this does not represent well $U_{\text{disk}}$. This is due to the complex inflow conditions particularly faced by the downstream turbines (T1, T2, ..., T80), with both axial and lateral velocity gradients affecting the relationship between the measured wind speed and $U_{\text{disk}}$. Additionally, for skewed wind directions, the measurement location might be in wake, while the rotor is not, or the rotor might be partially in wake, further decreasing the correlation between the power output and the measured wind speed.

When applying Eq. (1), as shown in Fig. 10-(b), the corrected wind speed is highly correlated with the power output, as the correlation between $U_{\text{disk}}$ and $P$ is not affected by the complexity of the flow field in the model. When further correcting with Eq. (2), as shown in Fig. 10-(c), the scatter in the power curve is not further decreased, and a shift towards slightly higher wind speed values is observed due to the correction of the turbine blockage. The corrected waked power curve values are compared with freestream power curve values $(U_\infty, P^I)$, i.e. power values obtained from isolated-turbine simulations are plotted against the wind speed retrieved at the turbine location from the freestream simulations (black squares in Fig. 10). We notice an overestimation of the power performance when correcting with Eq. (1) only, and strong agreement with all the turbines' power output when further correcting with Eq. (2).

Figure 10 includes freestream power curve values derived from an additional set of isolated and freestream simulations run at a hub-height wind speed of approximately 6.5 m/s, i.e. around 10% lower than the freestream velocity in the wind farm case. As shown in Fig. 10-(c), after applying the correction, the power curve obtained under waked conditions agrees with the freestream power curve values $(U_\infty, P^I)$ obtained at lower freestream wind speeds. Such agreement confirms that, under the

simulated conditions, the $(U_{\text{mast}}/U_{\text{disk}})^I$ ratio can be assumed as constant over the plateau of the thrust-coefficient curve for wind speed variations up to at least 10%.

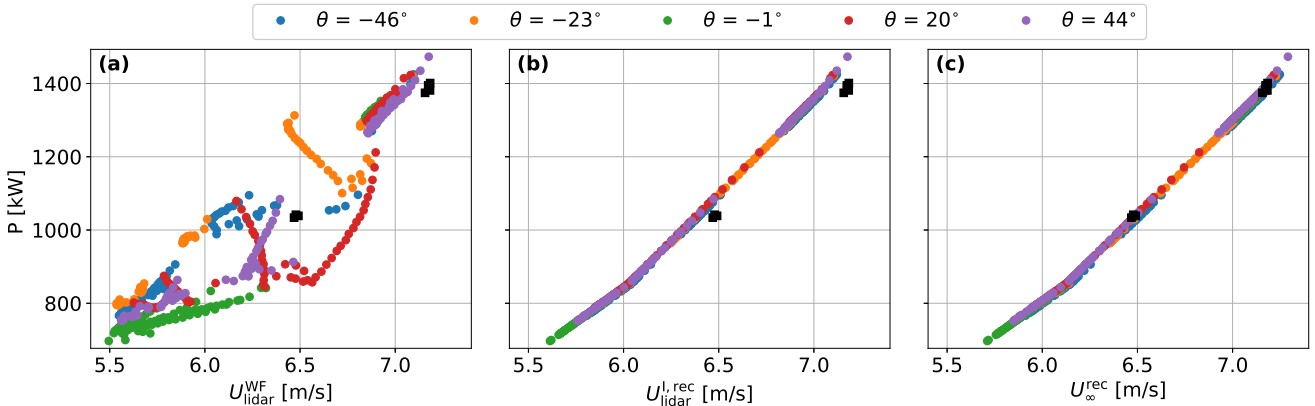

**Figure 10.** Power output against the lidar-measured wind speed of all the wind-farm turbines for all wind directions. $(a)$: no correction applied on the measured wind speed. $(b)$: wind speed corrected with Eq. (1). $(c)$: wind speed corrected with Eq. (2). Black squares indicate points with the combination $(U_\infty, P^I)$ from the isolated and freestream simulations.

The correction method also works well when replacing the term $U_{\text{disk}}$ in Eq. (1) with $U_{\text{disk,lidar}}$ retrieved at 0.5D in front of the rotors, as shown in Fig. 11. Although the scatter is slightly larger than when using $U_{\text{disk}}$, all lidar configurations allow for large improvements in the power curve. The results for the three commercial lidars in Fig.11-(a,b,c) are quite similar with no significant improvements when increasing the number of beams, while keeping the same opening angle ($\varphi = 15°, 18°$). However, when increasing $\varphi$ to $37°$, the correction results in significant less scatter, as shown in Fig. 11-(d). This suggests that $U_{\text{disk,lidar}}$ provides a better estimation of $U_{\text{disk}}$ when increasing the scanned area.

The short-range lidar measurements at 0.5D in front of the rotor do not provide an accurate estimation of $U_{\text{disk}}$. However, the measurements at 0.5D can be used to apply the correction method as they are highly correlated with the velocity at the disk ($U_{\text{disk}}$), as shown in Fig. 12. Since the lidar measures very close to the rotor, the correlation between $U_{\text{disk,lidar}}$ and $U_{\text{disk}}$ is not greatly impacted by the velocity gradients in the wake, and measurements and rotors might be both either inside or outside the wake. In agreement with the results in Fig. 11, $U_{\text{disk,lidar}}$ estimated from the circular scanning lidar measurements with $\varphi = 37°$ shows the highest correlation with $U_{\text{disk}}$ with a coefficient of determination $R^2 = 0.998$.

## 7 Discussion

### 7.1 Power curve definition

As described in the introduction, manufacturer-issued theoretical (MIT) power curves play a central role in both energy yield analysis (EYA) and power curve verification (PCV). In energy yield analyses, MIT power curves are commonly considered

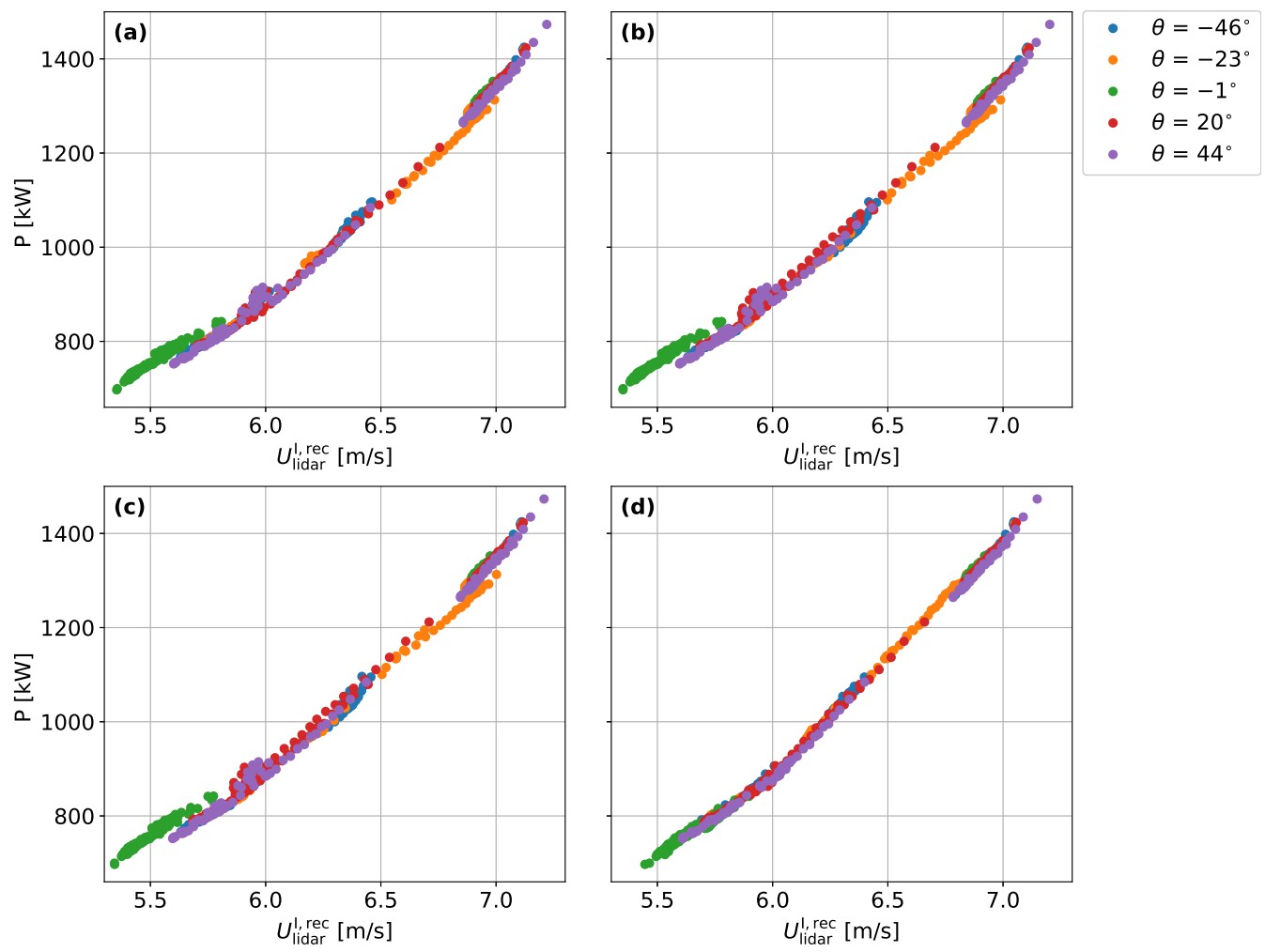

**Figure 11.** Power output against the lidar-measured wind speed of all the wind-farm turbines and all wind directions. Wind speed measurements are corrected with Eq. (1), where $U_{\text{disk}}$ is replaced with the term $U_{\text{disk,lidar}}$, which is estimated using measurements at 0.5D from the 2-beam lidar ($a$), the 4-beam lidar ($b$), the 50-beam ($c$) and the 50-beam ideal lidar ($d$).

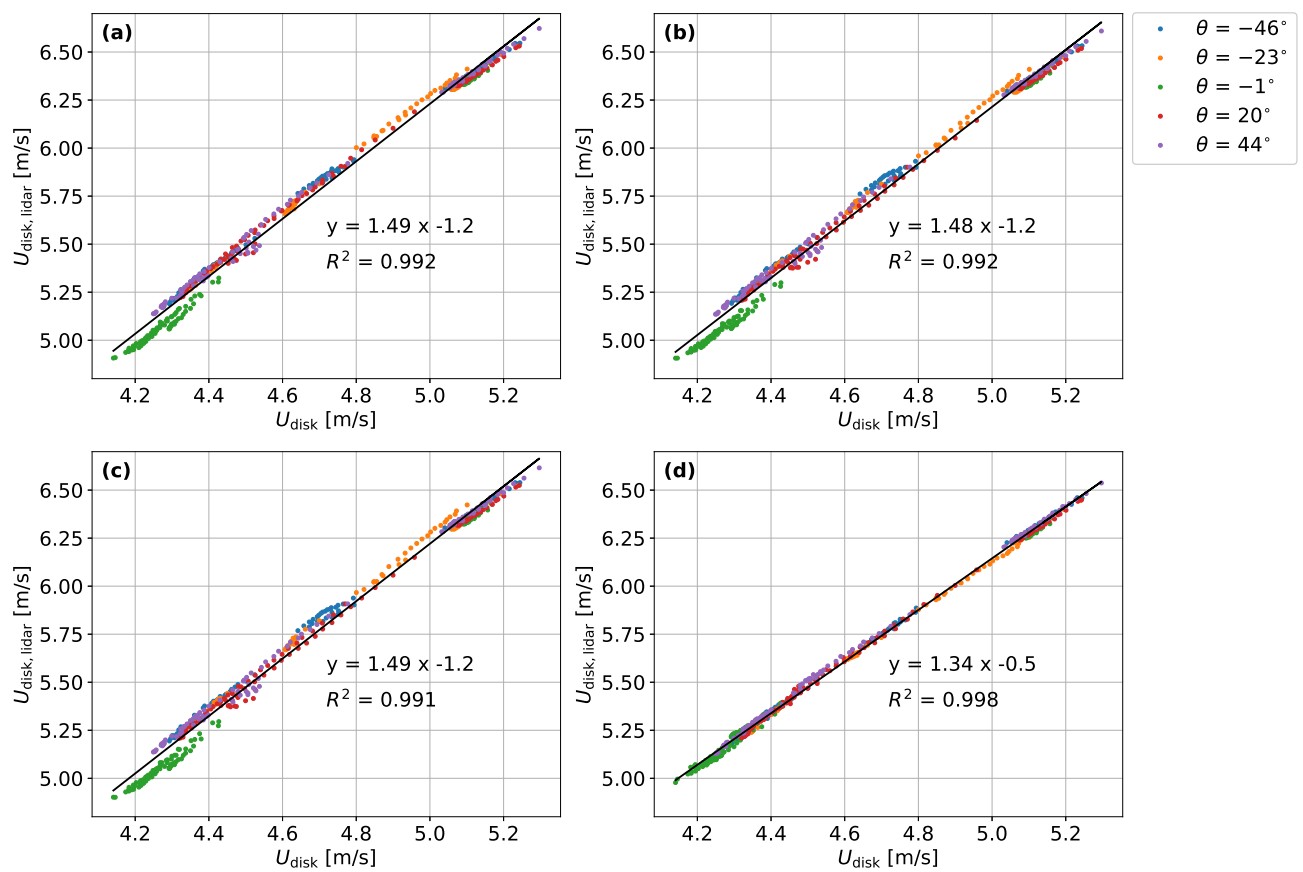

**Figure 12.** Scatter plots and related linear regressions between $U_{\mathrm{disk}}$ and $U_{\mathrm{disk,lidar}}$ estimated with measurements at 0.5D from the 2-beam lidar ($a$), the 4-beam lidar ($b$), the CSL ($c$) and the 50-beam lidar ($d$).

functions of *freestream* wind speed. Industry experience with PCV, however, suggests a different definition for this wind speed. Since MIT power curves are on average reasonably consistent with IEC-compliant power curve measurements (Harman, 2012), there is a strong argument to be made that MIT power curves, in effect, represent power as a function of *measured* wind speed.

How the correction methods proposed in this paper should be used depends upon the precise definition of the manufacturer-provided power curve. Of course, if one wants to use a measured power curve directly within an EYA, then the path is clear: just correct the curve using Eq. (1) and Eq. (2) as described herein. However, measured power curves are not commonly used in EYAs. The primarily use of a measured power curve is in PCV. For this application, the MIT curve needs to be precisely defined. Table 1 summarizes the correction implications for different MIT curve definitions.

The first row in Table 1 represents the most straightforward scenario, where power in the MIT curve is defined as a function of freestream wind speed for the turbine operating in isolation. In this case, an apples-to-apples comparison between the measured curve and the MIT curve requires correcting the measured curve using Eq. (1) and Eq. (2) as described in Sect. 2.

| MIT power curve definition | Correction to measured curve needed before comparing with MIT curve for PCV | Correction to MIT curve needed before use in EYA |
|---|---|---|
| Power of the turbine in isolation as a function of freestream wind speed | Eq. (1) and Eq. (2) | None |
| Power as a function of measured wind speed as it would be measured in the actual power curve test | None | Eq. (1) and Eq. (2) |
| Power as a function of measured wind speed as it would be measured in a reference test configuration for an isolated turbine | Eq. (1), but with an additional correction to get to mast/lidar wind speed in the reference test configuration | Eq. (2), but from mast/ lidar wind speed in the reference test configuration to freestream |

**Table 1.** Power curve correction methods for different definition of the manufacturer-issued theoretical (MIT) power curve. The power curve measurement is assumed to take place in a wind farm.

Alternatively, power in the MIT curve could be defined as a function of the wind speed as would be measured in the specific power performance test under consideration. For this case (second row of Tab. 1), no correction is required when comparing the measured curve with the MIT curve; however, Eq. (1) and Eq. (2) must be applied to the MIT curve before use in an EYA. A major drawback to defining the MIT curve in this way is that it implies a different MIT curve for different test configurations, as the measured power performance is sensitive to the relative locations of the test turbine and wind speed measurement as well as the locations of other wind farm turbines. A potential workaround is to define the power in the MIT curve as a function of wind speed as would be measured in a reference test configuration for an isolated turbine (e.g. wind speed measured 2D directly upstream). In such a case, an apples-to-apples comparison between the measured curve and the MIT curve would not require application of Eq. (2) to the measured curve, but just Eq. (1), along with an additional correction to get the wind speed that would be measured in the reference test configuration. Before using the MIT curve in an EYA, Eq. (2) would need to be applied, but in this case by correcting the wind speed that would be measured in the reference test configuration to freestream. The main drawback with this last definition, is the extra complexity involved in the corrections and the confusion that arises from having for each turbine model two types of theoretical power curves, one for EYAs and one for PCVs.

Clearly, the precise definition of a MIT power curve affects how the curve should be handled. Thus, the current situation where the definition of wind speed in these curves is ambiguous should be rectified. Based on the discussions herein, the authors recommend consistently and explicitly defining the power curve in the traditional way: power as a function of hub-height freestream wind speed for the turbine operating in isolation.

## 7.2 Practical application and limitations

Our results show that the correction method can potentially reduce both bias and uncertainty of power performance measurements. However, the approach relies on the accuracy of the flow model, which might introduce errors when applying the correction to field measurements. In addition, a large number of simulations may be required, given the potential sensitivity of $(U_{\mathrm{disk}}/U_{\mathrm{lidar}})^{\mathrm{WF}}$ to wind direction and the sensitivity of the correction factors to wind speed outside the constant-$C_T$ region of the $C_T$ curve. The computational expense could be mitigated through the use of engineering wind farm flow models and/or approximations to reduce the number of simulations required (e.g. assumptions about how the correction factors vary with $C_T$). Of course, the introduction of modelling simplifications implies a cost: added uncertainty in the calculated corrections.

The drawbacks of relying exclusively on numerical simulations to make the corrections could be mitigated by complementing the flow model with nacelle lidar measurements. Our numerical results indicate that short-range nacelle lidar measurements can be used to reduce the impact of turbine-induced flow disturbances on power performance measurements, which improves both accuracy and precision on the derived power curve. However, when using nacelle lidar measurements together with Eq. (1), the power performance would be still overestimated because of the difference between $U_{\mathrm{lidar}}^{\mathrm{I}}$ and $U_\infty$. In order to retrieve the freestream power curve $P = P(U_\infty)$, the lidar measurements must be further corrected with Eq. (2), which can only be used with the output from numerical simulations of both the isolated turbine and the freestream flow field. Thus, as shown in this work, nacelle lidar measurements can be used to correct for the effect of neighbouring turbines on the measured power performance, but simulations are needed in order to further correct for the blockage effect of the single isolated rotor.

During power performance tests of an isolated turbine, nacelle lidar measurements could be retrieved at both 0.5D and 2D in order to estimate $(U_{\mathrm{disk}}/U_{\mathrm{lidar}})^{\mathrm{I}}$. Then, when power performance tests of the same turbine model are carried out in a wind farm, the ratio $(U_{\mathrm{disk}}/U_{\mathrm{lidar}})^{\mathrm{WF}}$ can be retrieved with the same procedure and the measured power curve can be corrected with Eq. (1). However, $(U_{\mathrm{disk}}/U_{\mathrm{lidar}})^{\mathrm{I}}$ might be sensitive to site-specific effects, e.g., atmospheric stability conditions, which might be different at the isolated turbine location compared to at the wind farm site. Additionally, in case of different hub heights between the wind farm and the test site, ground clearance effects might cause variations of $(U_{\mathrm{disk}}/U_{\mathrm{lidar}})^{\mathrm{I}}$. In our numerical tests, both the isolated turbine and the wind farm operate under the same atmospheric conditions. This might improve the results compared to the case where measurements are obtained from different sites.

Numerical and experimental investigations (Meyer Forsting, 2017; Simley et al., 2016) showed that the induction factor $a = (U_\infty - U_{\mathrm{disk}})/U_\infty$ is not affected by moderate vertical velocity shear, while strong variations of both $a$ and $C_T$ have been observed under extreme vertical shear conditions with a power law exponent of 0.5 (Meyer Forsting et al., 2018). On the basis of such results, the variation of $(U_{\mathrm{disk}}/U_{\mathrm{lidar}})^{\mathrm{I}}$ among different sites is likely small when measuring under neutral or nearly-neutral conditions, while variations might be observed under stable conditions characterized by strong vertical shear. However, further investigation is needed to evaluate the sensitivity of the correction method to different vertical wind profiles and atmospheric conditions.

Also, the reliability of the correction method under waked conditions could be tested by conducting power performance measurements at the wind farm edge using nacelle lidar measurements. Depending on the wind direction, the reference turbine

would be either the most upwind or downwind of the farm. So, $(U_{\text{disk}}/U_{\text{lidar}})^{\text{I}}$ and $(U_{\text{disk}}/U_{\text{lidar}})^{\text{WF}}$ could be retrieved from wake-free and waked measurements, respectively. Equation (1) would be then applied to the waked measurements to evaluate whether they provide a power curve that is consistent with that obtained from the IEC-compliant wake-free measurements.

### 7.3 Opportunities for field validation of the numerical results

Comparisons with field observations can help us better understand the reliability of the proposed methodology and model-predicted correction factors. Although it is not possible to validate the correction factor predictions directly, there are some common types of field observations that can be used to validate model output that is relevant to the calculations in Eq. (1) and Eq. (2). For example, model output could be compared with nacelle-mounted lidar measurements of the streamwise variation of wind speed within the induction zone, providing an indication of how well the model is able to predict the wind speed relationship between the rotor disk and locations 2-4D upstream. Another approach could be to bin power curve measurements by direction to see how well the model is able to predict any observed performance variation with wind direction for a given wind speed.

Another indirect approach to validating the correction methodology and model predictions is to apply them to measured power curves. For example, the corrections could be applied to power curve measurements on a record-by-record basis to determine whether the correction methodology reduces scatter in the measured power curve. In addition, the waked vs wake-free comparison of corrected curves described earlier in Sect. 7.2 could also be used to evaluate model-based corrections. Finally, the methodology could be applied to power performance measurements taken at multiple turbines at an offshore wind farm or an onshore wind farm with flat terrain in order to investigate unexplained variations in the measured performance from turbine to turbine. In all these suggestions, the objective is to determine whether the correction methodology reduces variability in the measured performance.

## 8 Conclusions

In this work we present and evaluate a method to correct for the impact of turbine-induced flow disturbances on power performance measurements. The correction method is designed to recover the test turbine freestream power curve, i.e. a power curve that faithfully represents turbine power production as a function of freestream wind speed when it is operating in isolation. The method accounts for both the induction of the test turbine as well as the influence of surrounding turbines via blockage and sometimes wakes. Essentially, we take each wind speed value of the measured power curve and correct it to represent the freestream wind speed that would prevail if the test turbine were producing the same amount of power while operating in isolation.

Our CFD analysis suggests that the corrections can reduce uncertainty and bias in power performance measurements. Simulations of power performance measurements at a 100-turbine wind farm revealed variations in "measured" $C_P$ of more than 6% along the front row of turbines due to blockage effects. There was also a $C_P$ bias of 4.5%, primarily related to the impact of induction from the test turbine on the measured wind speed. The correction method was shown to reduce these sources of

variance and bias in the measured performance. The CFD analysis further demonstrated the potential to correct for much larger variances related to wakes.

The correction factors in this work derive from flow model output, and the same could be done when applying the methodology to real power performance measurements. That said, evidence from this study suggests that the correction factors, at least in part, could also derive from lidar measurements. Measurements taken just 0.5D upstream of the turbine are expected to be highly correlated with power output, much more so than 2D upstream, and the CFD analysis indicates that the correction factor related to the influence of surrounding turbines can be reasonably approximated using such near-turbine measurements.

Blockage effects appear to materially distort the outcome of IEC standard power performance measurements; reliable corrections to these effects would reduce uncertainty and produce curves that are more consistent with how power curves are defined and used in energy yield analyses.

The next step in this research should be to test the correction methodology on a set of real-world power performance measurements. Field observations could further clarify the validity and utility of these corrections. Additionally, it should be investigated how the method performs when CFD simulations are replaced with engineering wake models, which require lower computational costs and are more extensively used in energy yield analyses.

*Code and data availability.* Code and data related to this work might be obtained by contacting the authors.

*Author contributions.* AS, JB and AP participated in the conceptualization and design of the work. AS, JB and AP participated to regular discussions on the interpretation of the results. JB theoretically defined the correction method and performed the CFD simulations. AS defined the lidar-based application of the method and conducted the post-process analysis of the CFD results, retrieving the virtual lidar measurements. AS wrote the draft manuscript. JB and AP supported the whole analysis and reviewed and edited the manuscript.

*Competing interests.* The authors declare that they have no conflict of interest.

*Acknowledgements.* This work has received funding from the European Union Horizon 2020 through the Innovation Training Network Marie Skłodowska-Curie Actions: Lidar Knowledge Europe (LIKE) [grant number 858358].

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
