# Peer review of "A method to correct for the effect of blockage and wakes on power performance measurements"

_Wind Energy Science, 2023_

## Referee Comment (RC2)

General comments

The manuscript entitled "A method to correct for the effect of blockage and wakes on power performance measurements" describes a strategy to account for blockage when a turbine is operating within a wind farm rather than in isolation. The work is interesting and the motivation is strong, as blockage effects can have a large impact on power performance. However, the proposed method suffers from limited practicality, as either multiple simulations or lidar measurements are required to apply it to a real wind farm. While I recommend this work for publication in *Wind Energy Science*, I suggest the authors propose some solution to these limitations (e.g., using simplified models or SCADA measurements) either within the body of the manuscript or as a direction for future research.

Specific comments

1. Page 2, lines 50-51: This sentence in the introduction is unclear to me: "Specifically, wind farm blockage appears to affect the wind speed relationship between the mast location and the rotor in these results." What is the relationship and how is it affected?
2. Page 4, lines 106-107: Could you provide some support for the assumption that the blockage/induction remains constant with wind speed over the plateau of the thrust coefficient curve?
3. Page 5, lines 143-144: Can you comment on the effects of wind farm size and spacing?
4. Page 9, lines 215-217: Why do the largest power losses occur at the edges of the farm rather than at the location of the largest velocity deficits?
5. Page 13, section 5: Can you please elaborate on how this section ties into the rest of the manuscript?
6. Page 16, lines 324-327: Nacelle lidar measurements are also not trivial to obtain. Most turbines are still not equipped with lidars. Though it is interesting to propose measurements as an alternative to simulations, it does not significantly improve the practicality of this method.
7. Page 18, lines 336-337: Site-specific sensitivity is another strong limitation of this work. How much is this ratio expected to change under different conditions?

Technical corrections

1. Page 8, figure 3: Please put the axes in terms of *D*.
2. Page 11, figure 6: It is not clear how useful this plot is. The information shown seems redundant with figure 5. In addition, the variations shown are very small. If the authors choose to keep this figure, the y-axis should be labeled in terms of percent.
3. Page 12, figure 8: Once again, I am not sure how much this figure adds to the manuscript. The same information can be gleaned from figure 7.

4. Page 18, figure 14: This figure feels out of place here. It would make more sense to talk about the fidelity of the lidar measurements before talking about the relationship between lidar measurements and power (figure 13).

---

## Author Response (AR2)

Dear Nicolai Nygaard,

Thanks for your general comments on our work. We are thankful for your specific comments that certainly contribute to increase the value of the manuscript. Please, find our answers below. Comments from the reviewer are reported in bold and followed by our answers.

**The paper by Sebastiani, Bleeg and Peña addresses the influence of wind farm blockage on power performance measurements. They demonstrate through CFD simulations that met mast and nacelle lidar measurements of turbine power curves in a wind farm are affected in a material way by the blockage from the tested turbine and from the other turbines in the array. This introduces a bias in the measured power curve relative to the case of an isolated turbine. The authors propose and test a correction methodology, which can remove the effect of the wind farm and adjust the measured power curve to what would have been measured, if the turbine was operating in isolation.**

**They further introduce a second correction step, which removes the influence of the turbine's induction zone on the wind speed measurement. The result is a power curve where the power is a function of the freestream wind speed infinitely far from the turbine.**

**It is also demonstrated that the procedure can give meaningful power curve measurements in wakes.**

**Both corrections are in principle to be determined by numerical simulations, but the authors show through simulations of nacelle lidar measurements that measuring close to the rotor, 0.5 rotor diameters (D) upstream, can eliminate the need for a wind farm simulation. The remaining element of step 1 of the correction is the ratio of the wind speeds 2D upstream and at the rotor disk (or 0.5 D upstream if measured) for the turbine operating in isolation. This can be modelled, but the authors also suggest measuring this at the test site where the type certification is done using the same nacelle lidar setup to be used in the power curve verification at the wind farm.**

**Since power curve measurements do not generally find an underperformance relative to the manufacturer provided power curve, I would argue that the manufacturer's power curve is not specified at infinity, but at the distance range of 2-4 rotor diameters specified by the IEC standard, most likely 2.5D which is the recommended upstream distance in the standard. Likely, the power curves issued by manufacturers are generated to agree well with measurements taken at 2.5D.**

We agree with the raised concerns. The precise definition of hub-height "wind speed" in a manufacturer-issued theoretical (MIT) power curve is often not clear. Many assume this wind speed is a freestream wind speed, but the reviewer makes a strong case that it should be considered a measured wind speed. This difference in definition is relevant to the corrections proposed in this paper.

**I would therefore argue that correction step 1 (Eq. 1) is the only one we need to worry about for a power curve verification test (PCV). This test is meant to validate the power curve provided by the manufacturer and is typically done based on wind speed measurements at 2.5D.**

**As the authors show, these measurements are affected by the wind farm blockage. This will lead to a bias if this effect is ignored. Since the wind blockage tends to reduce the wind speed, this bias will make the power curve appear to be over-performing. Since the PCV is a contractual procedure linked to a performance warranty, this represents a commercial challenge.**

**The second correction step (Eq. 2) is necessary for EYA calculation where the freestream wind speed is the known characterization of the wind resource. For this a version of the power curve which depends on the freestream wind speed is needed. But it is not needed for PCV if the manufacturer's PC is given at 2.5D for a turbine in isolation, as I suspect. Rather, with this interpretation the second step is done not as part of a measurement campaign, but as a component of the EYA modelling and hence it can be accomplished using modelling tools.**

**I do not expect the authors to completely agree with the above, but I think there is sufficient uncertainty between these two views of what the manufacturer's power curve represents and hence the necessary steps needed to correct it that I wish for them to include a discussion of this in the paper. This is central to the corrections they devise and test in their work.**

We appreciate the reviewer raising this issue. If we defined wind speed in an MIT power curve as the reviewer has, we would agree with the reviewer on how the correction steps would need to be applied (though possibly with an extra factor to correct to/from the reference test configuration used to define the wind speed). The precise definition of wind speed in an MIT curve has implications for how the curve is used and how the corrections proposed in this paper are applied. We have added a detailed discussion of this matter to the revised paper.

**As a general remark why are the simulated measurements in the paper at a distance of 2D? The standard IEC recommends a distance of 2.5D and consequently most PCV campaigns use this distance to my knowledge.**

The standard recommends a distance of 2.5D, but IEC-compliant power performance tests can be conducted with measurements between 2D and 4D. Therefore, the chosen distance of 2D is in compliance with the IEC standard. The distance of 2D is chosen in order to measure as close as possible to the turbine, which might be the preferred choice when using nacelle-mounted lidars.

**Finally, I would like for the authors to comment on the use of engineering models for the single turbine and wind farm blockage as alternatives to the more expensive, but presumably more accurate CFD model. Have they tested a simpler model and compared the impact of blockage on the power curve with that from CFD?**

In principle, the authors consider it possible to apply the correction presented in this work with engineering models. It should be mentioned that uncertainty is likely to increase relative to using higher fidelity models. Testing the correction method with engineering models would be an interesting and meaningful extension of this work. These points have been added to the revised manuscript.

**Beyond the general comments above I have these specific suggestions:**

- **23 – maybe add "and warranted" to theoretical**

  Added in the revised version of the manuscript

- **L 27 – strictly speaking the freestream wind speed is measurable, just not concurrently with the turbine power output.**

  We agree and a small clarification is added in the revised version of the manuscript, specifying that the "concurrent freestream wind speed" is not measurable.

- **L 28 – consider changing "is expected" to "has traditionally been expected"**

  Text is modified according to the suggestion in the revised paper

- **L 38 – also cite:** https://web.archive.org/web/20140512231016/http://www.sgurrenergy.com/wp-content/uploads/2012/12/Compression-zone-technical-paper-A4.pdf

  The reference is added in the revised manuscript

- **L 50 – also cite https://iopscience.iop.org/article/10.1088/1742-6596/2265/2/022001 and discuss how and why the results in that paper differ from those in Sebastiani 2022**

The suggested reference [1] is added in the revised version of the manuscript, as we agree that, similarly to the other cited references at L 50, it is related to the topics discussed in this work. However, a discussion about the difference between the results from [2] and [1] is considered as out of the scope of the manuscript.

- **L 53 – also cite https://wes.copernicus.org/articles/6/521/2021/**

  We decided not to cite the Schneemann's paper as it is not really relevant for power performance tests. Schneemann et al. [3] showed results at transition piece height and at large distances from the turbine (between 10D and 40D) with the main goal of proving the existence of global blockage defined as the interaction between the atmospheric boundary layer and the whole wind farm.

- **L 65 - suggest changing "nacelle lidar measurements" to "simulated nacelle lidar measurements"**

Here the message is not that we are exploring whether the correction could be applied using simulated nacelle lidar measurements, but rather that we use simulated nacelle lidar measurements to investigate whether the correction could be applied with nacelle lidar measurements. Text has been modified in the revised manuscript to increase clarity.

- **L 99 – When using the average axial velocity across the rotor as a proxy for U_disk you are implicitly neglecting variations of the wind direction across the rotor and the effect that these might have on the power performance. Maybe comment on that.**

The authors acknowledge that a different definition of a power-equivalent wind speed could be used. It could be more comprehensive, for example, including factors such as how the wind vector varies over the rotor plane. The power-equivalent wind speed should be consistent with how the rotor forces are determined in the model. A more sophisticated rotor model along with a more sophisticated power-equivalent wind speed could lead to a different ratios in equation 1. However, our expectation is that the product of these two ratios probably will not change substantially.

- **L 129 – As mentioned above I do not believe that the PC and thrust curve is provided by the manufacturer as a function of the freestream wind speed, but regardless they need to be reformulated in terms of U_disk. The correction will just be smaller**

  The authors see your point regarding the power curve, but not entirely for the thrust curves, which are defined with blade element momentum models, which provide the thrust curve as a function of freestream wind speed. Thrust, rather than torque, is the dominant influence on this conversion, so assuming it is a function of freestream wind speed is the way to go. Also, the thrust coefficient invariably uses freestream wind speed in the normalization.

- **L 144 - What is the turbine rating and hub height? Include a plot of the power curve**

  Information regarding hub height (98 m) and rated power (3.45 MW) is included in the revised manuscript. The power curve cannot be shared due to proprietary reasons.

- **L 144 - Comment on how typical these spacing are. For offshore at least 3D is a very small spacing. Is it representative to have these spacings in a wind farm with 100 WTGs? What motivated this choice?**

  The size of the wind farm and the lateral spacing between turbines is chosen according to layout of wind farms built at sites with uni-directional or bi-directional wind roses, such as in the US Great Plains, where the wind roses are largely bi-directional.

- **Figure 2 – I propose to indicate rotor upper tip, lower tip and hub height**

  This information is added in the revised manuscript.

- **L 181 - The assumption of horizontal homogeneity is violated in the turbine induction zone, but you only need to assume symmetry, ie that the lidar beams measure the same wind vector. This is nearly true if shear and veer are neglected and the rotor and hence the lidar is perfectly aligned to the wind direction. Please include comments on this simplification**

  We agree with the comment. The assumption of horizontal homogeneity is violated within the induction zone as clearly shown in Fig. 10. However, it is reasonable to assume homogeneity between the two lidar measurements retrieved at hub height and at the same distance from the rotor, as shown, again, in Fig. 10. This clarification is included in the revised manuscript.

- **L 186 – You do not simulate the lidar probe volume. Please include an estimate of the size of this effect given the axial wind speed variation in the induction zone, the probe length and the measurement distance. Based on a rough estimate, is it reasonable to neglect it?**

  A shown in [4] and [5], typical pulsed lidars (the most indicated lidar to apply the correction method for the simulated turbine dimensions) have most of the probe volume in-between ±20 m around the measurement location. For the simulated turbine dimensions, this means around ± 0.15D. Such probe length is not expected to cause significant bias in conjunction with induction-induced wind speed gradients. Additionally, the impact of lidar probe volume and turbine induction was shown as negligible from previous works, which used lidars to retrieve accurate wind speed and turbulence measurements from inside the induction zone ([6], [7]).

- **L 190 – You neglect the vertical and lateral wind components. How accurate is this assumption? Why is it justified?**

  The vertical and lateral wind components are neglected only for the lidar measurements at 0.5D taken with the 50-beam and 4-beam lidars. Here the objective is not to provide an accurate estimation of the wind speed, but rather to measure a velocity quantity that is more highly correlated with $U_{disk}$ than the hub-height wind speed measured at 2D. Therefore, we do not care about retrieving an accurate estimation of the wind flow, as long as our measurement is highly correlated with $U_{disk}$, as shown in Fig. 14.

  For the lidar measurements taken with the 2-beam lidar (at both 0.5D and 2D), only the vertical wind component is neglected, which is a reliable assumption since the lidar is scanning horizontally (the third component of the line-of-sight direction vector is zero). In the real world, the lidar would not scan perfectly horizontally due to the turbine tilt, but the impact of the vertical wind component on the radial velocity would probably still be negligible, at least in simple flat terrain.

- **L 192 – "When using the 2-beam lidar focused at 0.5D, the horizontal wind speed at hub height is used as Udisk,lidar" but the wind speed at hub height varies laterally due to the blockage, so what have you actually done? Which wind speed are you using?**

  The horizontal wind speed at hub height refers to the horizontal wind speed reconstructed from the two radial velocities using Eq. (3). This information is added in the revised manuscript to increase clarity.

- **L 199 – Make it clearer that the standard deviation is the std over the 9 mast locations**

  Added in the revised version.

- **L 203 - Contrast the theta=-1 results with those of the single row in Sebastiani et al (2022) and in Bleeg and Montavon. Why are the Sebastiani results opposite those presented here and in Bleeg & Montavon for the wind direction perpendicular to**

**the row? Is it because Sebastiani et al did not include the effect of the boundary layer?**

Results from [2] are obtained for a single row of five wind turbines operating in uniform inflow (no boundary layer). Here, we investigate the case of a large wind farm, i.e., 5 rows of 20 turbines, operating within a conventionally neutral boundary layer. Therefore, the comparison between our results and those of [2] would be inconsistent due to both the strongly different layout and the different inflow conditions.  The comparison between the results from [2] and [1] is regarded as out of the scope of this work, though section 3.4 of [1] indicates that the inclusion of the boundary layer in the model, particularly the shear profile, likely contributes the difference in results between the two papers.

- **L 210 – I suggest adding "and hence the error bars" at the end of the sentence**

   We agree the comment increases clarity and it is added in the revised manuscript.

- **L 214 – why would you expect the power to be related to the wind speed raised to the power 2-3 and not simply to the power of 3? If you are trying to be more accurate and include the aerodynamic efficiency of the rotor as captures in the C_P curve, then I suggest adding a bit more detail. As it reads now it is simply confusing to anybody expecting the U^3 relationship. The details are also not important for the argument.**

   We agree with the comment and the sentence has been simplified in the revised version to avoid confusion, stating that the power output is related to the velocity to the power of 3.

- **L 218 – Is the reduced correlation due to the separation between the mast and the rotor in a turbulent (fluctuating) wind field?**

Turbulence would reduce the correlation, but that is not the reason for the imperfect correlation in the model. Power is function of the disk-averaged wind speed at the turbine. Blockage, whether from neighboring turbines or the test turbine itself, can change U_mast relative to U_disk. Therefore, the main source of the reduction in correlation is the separation between the mast and the rotor in conjunction with horizontal gradients induced by both the turbine and the whole farm. That said, we agree that turbulence would further decrease the correlation, especially when evaluating wind characteristics over 10-min intervals.

- **L 219 – What does Figure 5 show? Mean power over front row WTGs or the sum of their power or a dot for every turbine and wind direction?**

   A dot for every turbine, wind direction and mast location shown in fig. 1. That means a total of 900 dots (20 turbines, 5 wind directions, 9 masts). We added additional text to the revised manuscript to improve clarity.

- **L 258 – Again, this is up for discussion**

This discussion is more extensively covered in the revised version of the manuscript.

- **L 285 – I think the black squares near 7.1 m/s in Figure 12 represent the 20 front row turbines. This should be made clear in the text and in the figure caption.**

The black squares of Fig. 12 represent points from a "true" free-stream power curve, where the power output obtained from the simulation of the isolated turbine is plotted against the wind speed retrieved at the turbine location from the free-stream simulations, i.e., simulation with the same input conditions with no actuator disks. This was already mentioned at line 296, but additional text is added to increase clarity in the revised version.

- **L 287 - In wake the wind speed is highly inhomogeneous, and the two beams will see different radial wind speeds. The lidar will interpret this in the wind field reconstruction as a misalignment of the nacelle with the wind direction since it assumes a homogeneous wind speed. Is this not relevant because you are using the horizontal velocity = sqrt(ux^2+uy^2) and not the axial wind speed?**

This is true. Because of the inhomogeneity of the flow under waked conditions, the wind speed can be different at the two beam locations, causing large errors in the reconstruction of the single wind speed components (u, v), which are then smoothed by considering the horizontal velocity $\sqrt{u^2 + v^2}$.

- **L 320 – "However, the approach relies on the accuracy of the flow model ". I would add that it is not desirable to mix measurements and modelling for the purpose of a contractual PCV since this makes the resulting PC sensitive to details of the modelling setup. This is makes it much harder to define the PCV test in a contract and similarly challenging to define the uncertainty of the resulting power curve.**

  This is a good point raised by the reviewer, which is more extensively discussed in Sect. 7 of the revised manuscript.

- **L 356 - Typically the hub heights will differ between the test site and the wind farm. This would further challenge the re-use of (Udisk/Ulidar)^I since this will be affected by ground clearance effects on the flow.**

This is definitely something to consider in the application of the method using field measurements. We added this point to the discussion in the revised manuscript.

- **L 369 – Through the entire paper the reference distance is 2D, why is it 2.5D here?**

This was a mistake, the intention was to refer to the IEC-compliant measurements simulated through all the work, i.e., at 2D. This is changed in the revised manuscript, referring to 2D in the conclusions.

- **L 374 – "Field observations could further clarify the validity and utility of these corrections." Be specific about how this would be done**.

This discussion is extended in the revised manuscript.

**References**

[1]     C. Bleeg, J and Montavon, "Blockage effects in a single row of wind turbines," *J. Phys. Conf. Ser.*, vol. 2265, p. 22001, 2022, doi: 10.1088/1742-6596/2265/2/022001.

[2]     A. Sebastiani, A. Peña, N. Troldborg, and A. Meyer Forsting, "Evaluation of the global-blockage effect on power performance through simulations and measurements," *Wind Energy Sci.*, vol. 7, no. 2, pp. 875–886, 2022, doi: 10.5194/wes-7-875-2022.

[3]     J. Schneemann, F. Theuer, A. Rott, M. Dörenkämper, and M. Kühn, "Offshore wind farm global blockage measured with scanning lidar," *Wind Energ. Sci*, vol. 6, pp. 521–538, 2021, doi: 10.5194/wes-6-521-2021.

[4]     A. R. Meyer Forsting, N. Troldborg, and A. Borraccino, "Modelling lidar volume-averaging and its significance to wind turbine wake measurements," in *Journal of Physics: Conference Series*, Jun. 2017, vol. 854, no. 1, doi: 10.1088/1742-6596/854/1/012014.

[5]     W. Fu, A. Sebastiani, A. Peña, and J. Mann, "Dependence of turbulence estimations on nacelle lidar scanning strategies," *Wind Energy Sci.*, vol. 8, no. 5, pp. 677–690, May 2023, doi: 10.5194/wes-8-677-2023.

[6]     W. ; Fu, A. ; Peña, and J. Mann, "Turbulence statistics from three different nacelle lidars," *Wind Energ. Sci*, vol. 7, no. 2, pp. 831–848, 2022, doi: 10.5194/wes-7-831-2022.

[7]     A. R. Meyer Forsting, N. Troldborg, J. P. Murcia Leon, A. Sathe, N. Angelou, and A. Vignaroli, "Validation of a CFD model with a synchronized triple-lidar system in the wind turbine induction zone," *Wind Energy*, vol. 20, no. 8, pp. 1481–1498, 2017, doi: 10.1002/we.2103.

Dear referee,

Thanks for your general comments on our work. We are glad that you find it as interesting and strongly motivated. We are also thankful for your specific comments that certainly contribute to increase the value of the manuscript. Please, find our answers below. Comments from the reviewer are reported in bold and followed by our answers.

**General comments**

**The manuscript entitled "A method to correct for the effect of blockage and wakes on power performance measurements" describes a strategy to account for blockage when a turbine is operating within a wind farm rather than in isolation. The work is interesting and the motivation is strong, as blockage effects can have a large impact on power performance. However, the proposed method suffers from limited practicality, as either multiple simulations or lidar measurements are required to apply it to a real wind farm. While I recommend this work for publication in Wind Energy Science, I suggest the authors propose some solution to these limitations (e.g., using simplified models or SCADA measurements) either within the body of the manuscript or as a direction for future research.**

We agree with the reviewer upon the method requiring additional effort (either in the form of numerical simulation or short-range nacelle lidar measurements) compared to the current IEC standard. There are practical challenges, but we do not consider the applicability of the method to be poor. Multiple simulations are already conducted for calculating turbine interaction losses and energy production. In addition to that, the correcting method presented in this work only requires the simulation of a limited range of wind conditions and wind directions. Additionally, nacelle-mounted lidars are already a standard instrument for power performance tests, especially offshore. The additional measurements at 0.5D required for the method will not be a practical issue as most commercial lidars provide simultaneous measurements at different heights.  Nevertheless, following the reviewer's suggestion, in the revised manuscript, we further discuss the applicability of the method and the expected direction of future studies including the evaluation of engineering wake models.

**Specific comments**

**1. Page 2, lines 50-51: This sentence in the introduction is unclear to me: "Specifically, wind farm blockage appears to affect the wind speed relationship between the mast location and the rotor in these results." What is the relationship and how is it affected?**

We agree that the sentence as written is not clear and will be changed in the revised manuscript to "Specifically, wind farm blockage appears to affect the ratio between the wind speeds at the mast location and the rotor".

**2. Page 4, lines 106-107: Could you provide some support for the assumption that the blockage/induction remains constant with wind speed over the plateau of the thrust coefficient curve?**

We state that the induction is constant over the plateau of the thrust coefficient for small changes in wind speed. This assumption is reasonable and further supported by the results of Fig.12, where it is shown how the correction is still accurate with a decrease of 0.5 m/s in wind speed.

**T3. Page 5, lines 143-144: Can you comment on the effects of wind farm size and spacing?**

There would probably be sensitivity to both wind farm size and spacing. It is not easy to estimate this without running some sensitivity studies, though higher spacing and less turbines would likely cause a reduction of the wind-farm-related correction factors. However, previous studies suggest that wind-farm-related correction factors would not disappear even in the simplest case of a single row of turbines, as it was shown that the power performance at such site is different from that of a single isolated turbine.

**4. Page 9, lines 215-217: Why do the largest power losses occur at the edges of the farm rather than at the location of the largest velocity deficits?**

This is due to the poor correlation between the power output and the wind speed retrieved at 2D from the rotor. The power deviations of Fig. 4-c are consequence of variations among the U_disk of the different rotor, while fig. 4-a and 4-b shows deviations in wind speed measurements retrieved at 2D in front of the rotors.

**5. Page 13, section 5: Can you please elaborate on how this section ties into the rest of the manuscript?**

A few introductory lines have been included in the beginning of the section in the revised manuscript.

**6. Page 16, lines 324-327: Nacelle lidar measurements are also not trivial to obtain. Most turbines are still not equipped with lidars. Though it is interesting to propose measurements as an alternative to simulations, it does not significantly improve the practicality of this method.**

We touched upon this point in our answer to the general comment.

**7. Page 18, lines 336-337: Site-specific sensitivity is another strong limitation of this work. How much is this ratio expected to change under different conditions?**

As mentioned in the manuscript, the possible variation of this ratio for different sites is a concern to investigate in the future extensions of this work. This could be done by looking at specific turbulence and stability conditions to characterize their effect on the ratio. However, we would like to point out that short-range lidar measurements would allow for the best correlation with the "disk" velocity independently of the site-specific conditions.

**Technical corrections**

**1. Page 8, figure 3: Please put the axes in terms of D.** Changed as suggested in the revised manuscript.

**2. Page 11, figure 6: It is not clear how useful this plot is. The information shown seems redundant with figure 5. In addition, the variations shown are very small. If the authors choose to keep this figure, the y-axis should be labeled in terms of percent.**

**3. Page 12, figure 8: Once again, I am not sure how much this figure adds to the manuscript. The same information can be gleaned from figure 7. 4.**

We followed the reviewer's suggestion and removed both Fig. 6 and Fig. 8 from the revised manuscript.

**4. Page 18, figure 14: The figure feels out of place here. It would make more sense to talk about the fidelity of the lidar measurements before talking about the relationship between lidar measurements and power (figure 13).**

We decided not to change the order of the figures, as Fig. 14 provides the explanation for what we observe in Fig. 13, i.e. the good performance of the correcting method using the short-range nacelle lidar measurements (Fig. 13) is explained by the good correlation of such measurements with U_disk (Fig. 14). Therefore, we prefer to choose Fig. 13 before Fig. 14.